# Education for Environmental Sustainability and the Emotions: Implications for Educational Practice

Lynda Dunlop [1] and Elizabeth A. C. Rushton [2,*]

1   Department of Education, University of York, Heslington YO10 5DD, UK; lynda.dunlop@york.ac.uk
2   School of Education, Communication and Society, King's College London, London SE1 9HN, UK
*   Correspondence: elizabeth.rushton@kcl.ac.uk

**Abstract:** Increasing attention is being paid to the emotions in education and in communication about the climate crisis and other sustainability challenges. This has tended to focus on the relationship between emotions and environmental perceptions and behaviours. In this study, we understand emotions as evaluative feelings which meaningfully connect people and their environment. We draw on data from teachers, teacher educators, and young people (n = 223) to describe educationally-relevant emotions and identify the implications for educational practice. We argue that emotionally-responsive pedagogies are needed to identify responsibilities, develop coping potential, and improve future expectations. These pedagogies must act on the causes and consequences of environmental damage and develop teachers' and students' capabilities to take action and ultimately transform emotional appraisals. A more enabling policy environment is needed for teachers to adopt these approaches and empower them to take action relating to climate and ecological crises.

**Keywords:** capabilities; climate change; education; emotions; environmental sustainability; pedagogy; teacher educators; teachers; youth





## 1. Introduction

There is growing interest in the role of emotions in relation to environmental sustainability generally and climate change specifically. New threats to human health such as 'climate anxiety' are being defined and diagnosed [1] and have been found even where people do not have direct personal experience of climate change [2]. While there is growing interest in the links between emotions and environmental attitudes, intentions, and behaviours [3–6], it is important to avoid treating emotions (and particularly fear and hope) as levers to be pulled in order to achieve behavioural change [7]. In this work, we examine the place of emotions in education for environmental sustainability and the connection between emotions and actions. Throughout this article, we use the term 'climate change' to describe changes to the global climate system which are driven by anthropogenic activity, rather than changes which are induced by non-human changes or interventions.

Education has an important role to play in helping countries to meet the Sustainable Development Goals (SDGs) [8] and the goals of the Paris Agreement [9]. For example, SDG target 13.3 is to 'improve education, awareness-raising and human and institutional capacity on climate change mitigation, adaptation, impact reduction and early warning'. Indicators for success include the extent to which sustainability is mainstreamed into education policies, curricula, assessment, and teacher education [10]. These goals are consistent with youth calls to 'teach the future' and 'ensure that all students are substantively taught about the climate emergency and ecological crisis' [11], and with the role young people see education playing in responding to environmental crises [12]. However, these aspirations do not reflect the current reality in many jurisdictions, including England, where policy relating to sustainability and climate change education is absent [13], obstructive [14], and diminished [15]. The Department for Education's recent draft strategy for sustainability and

climate change education in the education and children's services systems [16] does not appear to meet calls from young people and teachers, and has been described as a 'placebo' policy', 'one which appears to do something, but fails to address the fundamental problem—leaving teachers and young people no better equipped to deal with the climate crisis' [17].

Although there is recognition that knowledge gain is insufficient for effective climate change education internationally [18], an emphasis on knowledge gain tends to dominate [19]. There is a need for education which helps young people to cope by 'listening and providing opportunities for active engagement . . . and [which] build[s] a sense of efficacy and a capacity to tackle the crisis and adapt to climate impacts' [20] (p. 201). Davidson and Kecinski [21] argue that attending to the emotions in education is crucial in supporting constructive responses to climate change. Similarly, Pickard [22] argues that the design of appropriate, emotionally-engaged pedagogies may be considered one dimension of 'generational responsibility' [22] whereby teachers play a role in ensuring that the next generation is left with at least as good a planet as the previous generation. Attending to the emotions in education is challenging, as in practice teachers report experiencing an 'emotional load' when teaching environmental and sustainability education [23], and anger, hopelessness, fear, and anxiety are prevalent climate change emotions among both teachers and young people [24,25]. Recent research has underlined the importance of understanding the full range of children's psychological responses to climate change, including positive and negative emotions, in order to promote both well-being and constructive engagement in education in relation to the issues of climate change and sustainability [26–29] In this study, we examine young people's emotions in education for environmental sustainability as they participate in research projects which aim to be interdisciplinary, creative, and affect-driven. We approach emotions as cognitive appraisals (as described in the following section) and identify implications for education based on the emotions expressed and their potential for transformation.

*Conceptual Framing*

Emotions can be understood as experiences, evaluations or motivations. This present work is situated in the evaluative tradition and treats emotions as cognitive appraisals [30], supported by evidence that appraisals cause emotions [31]. As Helm [32] argues: 'emotions are essentially affective modes of response to the ways our circumstances come to matter to us, and so they are ways of being pleased or pained by those circumstances' (p. 248). According to Lazarus [30], appraisal occurs at two levels: the first establishes the significance of the circumstances, and the second assesses the ability a person has to cope with the consequences of circumstances. The circumstance we are concerned with in this study is environmental unsustainability in an educational context, which includes emotions associated with climate change. Primary emotional appraisal depends on what is at stake and how important it is to the person, whether the circumstances are beneficial or harmful, and how this relates to the person's moral values or identity [30]. Secondary appraisal depends on the attribution of responsibility (such as credit or blame) and the extent to which the person thinks they can cope and that things will work out [30]. Cognitive appraisal models of emotions are not a universal way of understanding emotions: Ellsworth and Scherer [33] provide a review of related yet distinct conceptions of emotions. We argue that the cognitive appraisal model of emotions, with the incorporation of the appraisal of 'future expectancy', the extent to which a situation may improve or worsen, is a fundamental component of emotions related to coping with the climate and with ecological crises that are temporally and spatially complex.

To return to the example referred to in the introduction, anxiety is an emotion associated with the climate crisis [25,34,35], leading to symptoms such as panic attacks, insomnia, and obsessive thinking [1]. 'Climate anxiety' is increasingly prevalent as a term and has been used by a range of stakeholders, including policy-makers. Recent research has considered the ways climate change poses risks to mental health [36], including the impact of long-lasting adverse weather events such as floods and drought [37]. Drawing

on the description of Lazarus [30], Table 1 presents an analysis of anxiety in the appraisal tradition. Anxiety is associated with powerlessness, because there is no single agent to be held responsible. It can be distinguished from other emotions, for example anger (where blame is directed at known others), or shame and guilt (where blame or responsibility is directed at the self, and depends on whether it is possible to improve the circumstance). This analysis demonstrates the link between emotions and action, and therefore between the individual to their social and political context. We have chosen to highlight anxiety in Table 1, as this emotion is often the focus of emotions related to climate change, as demonstrated in our previous consideration of 'climate anxiety'. However, here we use this framing of emotions to investigate the emotions of young people and teachers in the broader context of education for environmental sustainability and to identify implications for pedagogy.

**Table 1.** Analysis of emotions based on Lazarus [30].

| | Appraisal Question | Anxiety |
|---|---|---|
| **Primary appraisal** | What is at stake and how important is it? | The self and others; very important. |
| | Is the circumstance harmful or threatening or beneficial? | Both harmful (in the present) and threatening (to the future). |
| | Does the goal at stake preserve or enhance a person's ego identity or moral value? | An existential threat is posed to the self and others now and future generations. |
| **Secondary appraisal** | Can responsibility be attributed—and to whom? | Responsibility cannot easily be attributed. |
| | Can the relationship between the person and the circumstance be improved? | Difficult to determine (because the threat is vague and spatially and temporally complex) |
| | Will things work out favourably? | Difficult to determine. |

## 2. Materials and Methods

### 2.1. Research Design

A qualitative participatory research design was used in this study. Following institutional ethical approval from Author 1's Research Ethics Committee (9 March 2020, Ref 20/18), we developed and ran two series of online participatory workshops for two separate but complementary projects, (1) *Geoengineering: A climate of uncertainty?* (*Geoengineering*) and (2) *Manifesto for Education for Environmental Sustainability* (*Manifesto*) (Table 2). The *Geoengineering* project foregrounded youth perspectives in decision-making about technological responses to climate change based on large-scale human intervention in the Earth's climate, also called 'geoengineering'. This understanding of geoengineering was the definition developed and shared by researchers and participants at the outset of the project. Participatory online workshops with young people from across Europe were facilitated by a team drawn from disciplines including education, philosophy, and policy. A key output of the workshops is a *Geoengineering Youth Guide and Policy Brief* [38–40]. During the workshops participants were able to learn about a range of geo-engineering approaches, including carbon geoengineering (such as ocean liming, ocean fertilisation, and carbon capture and storage) and solar geoengineering (including space mirrors and cloud thinning) [38]. The *Manifesto* project aimed to co-create an illustrated *Manifesto for Education for Environmental Sustainability* (EfES) for the four jurisdictions of the UK with young people (16–18 years) and teachers to articulate a shared vision of what the future of EfES could look like; the *Manifesto* was launched in November 2021 [41,42]. Both sets of workshops involved exercises to create intimacy, for example, by discussing which values participants prioritised in education for environmental sustainability (*Manifesto*) and identifying shared priorities, expectations, and worries (*Geoengineering*). The creation of intimacy has been found to support social cohesion and collective engagement and to lower

ratings of ostracism compared with situations where information is prioritised, differences attributed to the emotional tone of discussion [43].

**Table 2.** An overview of the online participatory workshops for the *Geoengineering* and *Manifesto* projects.

| Project | Events | Participants | Co-Produced Outputs |
|---|---|---|---|
| *Geo-engineering: A climate of uncertainty?* (*Geo-engineering*) | April–May 2021 4 × 5-h online workshops | Thirteen youths (18–25 years) from countries including: Albania, Belgium, Czech Republic, the Netherlands, Poland, Portugal, and the United Kingdom. | Youth Guide and Policy Brief [30,31] |
| *Manifesto for Education for Environmental Sustainability* (*Manifesto*) | May–June 2021 9 × 2-h online workshops | 210 participants from three groups: (i) youth aged 16–18 years (including those with declared Additional Educational Needs); (ii) teachers; and (iii) teacher educators from England, Scotland, Wales, and Northern Ireland | Illustrated Manifesto for Education for Environmental Sustainability [32] |

*2.2. Data Collection*

Across both projects, we hosted a total of thirteen workshops involving 223 participants. Following the completion of both series of workshops, we interviewed thirteen youth participants involved in the *Geoengineering* workshops. Voluntary informed written consent was obtained from participants prior to the workshops and from young people prior to interviews. We requested only personal information relevant to the workshops, namely, school, location, accessibility needs, and reason for participating for the *Manifesto* project; for the *Geoengineering* project, we requested nationality and age as well.

*2.3. Analysis Process*

Data were gathered from (1) recordings of workshops; (2) responses sent in from young people and teachers; (3) artefacts from workshops, including *Google jamboard*, *mentimeter*, *miro* and *padlet* posts, and *Zoom* chat; and (4) post-workshop interviews with young people. A conventional approach to qualitative content analysis [44] was used. In the first rounds of analysis, Author 2 coded data according to different emotions present in the data, for example, fear, anxiety, and hope. This process was completed and then the data set was reviewed two further times by Author 2 and once by Author 1 in order to ensure that all examples had been identified. In the second phase of coding and analysis, both authors discussed the clustered data set in order to explore the connections, nuances, and divergences across the different workshops and participants. We considered the data in the light of our own reflections as researchers and workshop facilitators. Our analysis was an interactive and discursive process, scaffolded by a shared *Google* document which enabled us to add reflections and questions; we continued to explore these together through conversations conducted via email and *Zoom*. The open and iterative approach to identifying themes, which included frequent reengagement with the data set, had the advantage of not imposing theoretical perspectives on the data and ensured that the conclusions we drew were an accurate reflection of the discussions and ideas shared by participants through the two series of workshops. Data are reported according to whether they were derived from a teacher (T), teacher educator (TE), or youth (Y) contribution from either the *Manifesto* or *Geoengineering* project.

**3. Findings: Emotions That Youth, Teachers and Teacher Educators Bring to Discussions of Education for Environmental Sustainability**

Here, we explore the emotions that youth, teachers and teacher educators share when discussing EfES. These include feelings of powerlessness and individual action alongside

systemic accountability as well as current anxieties juxtaposed with hope for the future. We begin by considering the ways in which participants shared their fear of judgement.

### 3.1. Fear of Judgement and Isolation

When articulating the barriers to EfES through the *Manifesto* project, youth described emotions such as 'shame', 'guilt', 'apathy', and 'fear'; teachers and teacher educators also shared these negative feelings, in addition to those of 'blame' and 'powerlessness'. Young people described their fear of judgement when trying to bring about change in the context of EfES as one young person described:

> I think ... there's a lot of fear of judgement as well, like nobody wants to make that first move, everyone's scared of what everyone else is going to say ... we're in central London and we wanted to make the ... we were on a residential street, our school is, an we said, "Oh, can we make that car-free access in the morning, so less people, like drive to school." And there was, like, so much hit back ... and nobody ever wanted to be actually the one to catalyse the discussion ... so yeah, I guess fear of being judged. (Youth, *Manifesto* project)

Fear of judgement connects with young people's description of EfES not being seen as 'cool' and a perceived lack of compassion and passion amongst their peers, which they say makes it difficult to act on their own empathy or compassion. Furthermore, fear of judgement is compounded by young people's experiences of not being able to bring about the change they would like to enact within their school communities. This was an experience shared by another young person who said:

> So where I was going with the whole idea of belittling ... like exacerbated, especially when, if you're in discussion with senior leadership staff at your school and you try and ask for, not radical change, but what they might think is radical change ... it is quite easy to think ... that you are ... asking for a bit too much, or if you're not belonging ... or your ideas aren't as good as they are, so I think that can then restrict further change because if ... people are suffering with imposter syndrome, they're not going to ever step out again ... if your school's response isn't as proactive as you'd like it to be, it really restricts you from ever asking for change again, because you think you are going to be met with the same response. (Youth, *Manifesto* project)

While fear can mobilise (to avoid or escape the emotion), we see here that if it is unlikely that mobilisation can happen or will lead to change, there is a risk of apathy. Several young people shared how they needed the help of the older generation and expressed their frustration at not being taken seriously by those older than them, which led them to feel isolated:

> I feel like in a way there's a bit of kind of, like, isolation for our younger generation with the older generations ... they kind of make you feel like we can't do anything ... I feel like there's, yeah, kind of isolation in certain groups that feel like they have more power in a way. So, like, distribution of authority and responsibility. (Youth, *Manifesto* project)

In contrast, other young people talked about feeling encouraged or hopeful that spaces such as this were being created to enable young people to work together; for example, a young person who participated in the *Geoengineering* project said, 'I feel so encouraged that there are so many people willing to work on something like this, share, learn, and so much expertise!'. And here, they contrast the action taken during the project with what is often missing from education:

> At the same time, being with people that at least try to learn about climate change, and about potential solutions, it's nice to see, but then, it's like, learning and talking, and it's acting, and I think the acting part is really missing. I'm not blaming individuals, but just in general. (Youth, *Geoengineering* project)

### 3.2. Fear of Powerlessness

Feelings of fear relating to taking action in relation to environmental sustainability are exacerbated when young people lack opportunities to participate in decision making or, in educational settings, perceive that their intentions and ideas are rebuffed or 'belittled' by adults in leadership positions in their school. This can lead to a sense of 'imposter syndrome' and powerlessness, which is directly associated with secondary appraisal (options and prospects for coping), in particular in assessing whether or not the situation can be improved.

Teachers recognise the feelings of powerlessness experienced by the young people they teach who have knowledge about environmental sustainability without the capacity to make decisions. One teacher shared a conversation they had with their 12-year-old daughter about climate change and the environment and reported that she said, 'Well, we learn quite a lot about this sort of stuff at primary school ... but we never got to actually do anything' (teacher, *Manifesto* project). Another teacher said of their pupils:

> *Sometimes they feel a little bit powerless, because they can't make decisions and that decisions are made by people further up ... they're [the pupils] saying they would like it to be sort of something that they get to do, rather than it just being on top of everything else that's expected of them.* (Teacher, *Manifesto* project)

Teachers themselves reported feelings of powerlessness linked to government policy which marginalises environmental sustainability and climate change in curricula and accountability frameworks, and noted that as a result it is given little space in teacher education and professional development. One teacher involved in the *Manifesto* project said, 'we can lack confidence because we are navigating this ourselves and don't feel like experts where we might [be] in our subject.' The connection between feelings and being able to act was identified as important by teachers; for example:

> *We offer the basic knowledge, and say they are inheriting a world that is not what it should be. However, we don't offer solutions and a platform for them to discuss and enact their thoughts/feelings.* (Teacher, *Manifesto* project)

When discussing how to respond to pupils' feelings of powerlessness, teachers highlighted the importance of incorporating opportunities for EfES from an early age and providing opportunities to 'see the difference that they can make just by doing little things. And that may inspire them to then go on to potentially do bigger things as they get older' (teacher, *Manifesto* project). This approach, of demonstrating how small individual actions can make a difference was one shared by young people as one youth participant commented:

> *I find a lot of time in our school is the mentality is that people feel like they can't do very much because they're a singular person, and I think encouraging people to think of themselves as someone who can make changes within a system and cause the whole system to eventually change is very ... is a way that, is something that needs to happen.* (Youth, *Manifesto* project)

Powerlessness was associated with the risk of apathy. Describing participation in protests against a dam, the following youth participant links powerlessness to questioning whether to carry on with their activism or to emotionally detach from the situation.

> *Not really, except to be persistent. It happens that okay, these doors shut and you feel defeated, and so people give up. It happened to me as well ... for example we had this problem with many hydropower plants which were really bad for ecosystems ... and actually that really helped. The whole protest really helped, but at one point you just feel exhausted, like, how many times can I go in the street and protest? How many times can I meet in the group and come up with concrete suggestions and send it to the policymakers or whatever? How many times can this be ignored that I feel well with myself, then I feel like it's all for nothing?* (Youth, *Geoengineering* project)

Present in this young person's response is both the sense of individual action building to systemic change; this was a feature of wider contributions shared as part of the *Manifesto* project.

### 3.3. Guilt, and Feelings of Pressure

Guilt involves a moral transgression, and is felt where blame is directed at oneself. Through their contributions to the *Manifesto* project, young people, teacher educators, and teachers all highlighted how they perceived the burden, blame, and responsibility of education for environmental sustainability to be felt by individuals, whereas accountability for action and change should be systemic. For example, one young person said, 'don't punish people who don't have the choice to be sustainable' and another agreed, 'punishment for people who have the option to be sustainable, but don't, and incentives for people to be sustainable to encourage it' (youth, *Manifesto* project). Similarly, in the *Geoengineering* project, young people described wanting 'empowering first, shaming later' (youth, *Geoengineering* project).

Rather than individuals, young people felt their governments and large companies should be accountable for implementing sustainable practises across society; as one young person said:

> *At the end of the day, I shouldn't feel guilty if I can't afford a bamboo toothbrush. Instead, the onus needs to be on governments to make sure companies are making the sustainable options the most affordable and accessible.* (Youth, *Manifesto* project)

The framing and terminology used to describe sustainability was occasionally problematic for young people in the way that it emphasised individual rather than collective responsibility, for example a young person commented 'things like the term 'carbon footprint' is just set up to put the blame on individuals alone' (Youth, *Manifesto* project). This idea of individual responsibility, and therefore guilt at not being able to realise EfES across a school community as a single enthusiastic teacher, resonated with teachers; one reflected:

> *I totally agree . . . it is all down to individual teachers who get it and have time/energy/sufficient guilt or anxiety to keep going at it which isn't sustainable and doesn't spread the burden.* (Teacher, *Manifesto* project)

The guilt and pressure of being accountable for EfES at an individual level was felt by several teachers to lead to exhaustion and burnout, as this conversation between three teachers demonstrates:

> *T2: Do you ever feel like you get worn down by doing more than your bit?...The idea you can [be] the most passionate person in the world and six months later they are a broken person . . .*
>
> *T3: I was going to say on that, I've stopped sorting out the bins and they've got so bad that I think they might bring in a proper system so, yeah, definitely sitting on your hands even though it is an emergency . . .*
>
> *T2: I used to do that. Wednesday afternoon me and the caretaker would do the recycling . . . .*
>
> *T3: Yeah, and it's quite exhausting like trying to keep people onboard and not get annoyed with them and not sort of think . . .*
>
> *T4: Yeah, not get annoyed at them.*
>
> *T3: Yeah, when they ask me how to recycle stuff I'm like, 'Just look it up yourself' but . . .*
>
> *T4: Yeah.*
>
> *T3: ...you have to be nice.*
>
> *T4: Yeah, bins are wearing.* (Teachers, *Manifesto* project)

Although this example of recycling bins may appear inconsequential, the teachers in this discussion use the lack of collective responsibility for recycling school waste as an analogy for the wider apathy of others within the school community and the willingness to burden a few passionate people with work that should be shared equally. Furthermore, those burdened few are expected to be 'nice' in their interactions with others when discussing environmentally sustainable actions. This sense of frustration is reflective of the tension

which young people expressed in relation to the inaction of some of the 'older generation' previously described.

Across the contributions made during both projects which incorporated emotions, participants shared their hopes for the future as well as their anxieties and worries; we now turn to explore these perspectives.

### 3.4. Anxiety

Participants shared their anxiety about environmental issues in general and climate anxiety specifically; for example a young person said, 'I have climate anxiety big time' (youth, *Geoengineering* project), and added:

> *I'm 17 and I know for me and for a lot of other people my age, environmental issues are really important to us and a definite source of anxiety. By increasing the amount of education and the quality of education about the environment, hopefully it would be beneficial to overall mental health as it could be seen as an issue to be improved rather than just having anxiety about it.* (Youth, *Manifesto* project)

Teachers and teacher educators highlighted the anxiety and lack of confidence that teachers felt in relation to environmental sustainability, expressing their concerns as follows:

> *I am worried about overwhelming pupils with the gravity of the situation, although I understand it is extremely important that they are taught about this.* (Teacher, *Manifesto* project)

> *A hard topic to face—a wicked problem -...and we do not have tools . . . teachers don't want to embed anxiety in students by trying to face up to the enormity of the problem—we don't feel trained to do this.* (Teacher educator, *Manifesto* project)

Anxiety was often coupled with hope for the future.

### 3.5. Hope for the Future

When describing what was needed for the future of education for environmental sustainability as part of the *Manifesto* project, young people wanted 'less ecoanxiety on the whole', 'better contentment and understanding', 'love the world we are in' and, 'my dream . . . a future full of colour'. While they feared the worst, they yearned for better. Similarly, teacher educators spoke of hope and values: one teacher educator asked, 'how do we develop pedagogies of hope' and another spoke of 'embedding in values'. Relatedly, one teacher drew a parallel between the COVID-19 global pandemic and the environmental crisis, saying, 'In the future . . . hopefully have a better mindset from covid (seeing a worldwide disaster happen)—hopefully can relate that to the environment' (teacher, *Manifesto* project).

One feature of a more hopeful future for the environment articulated across both projects was the desire to empower, rather than elicit negative feelings through threatening or shaming. For example, one person noted that 'negative feelings could automatically lead to conflict and instilling negative feelings might not therefore not be the most fruitful way to proceed' (youth, *Geoengineering* project). Similarly, when responding to the prompt 'The key message I want to communicate about geoengineering is . . . ', one young person said, 'climate change is a life limiting or terminal illness for us all now and in the future unless we all act' (trying to empower not shame) (youth, *Geoengineering project*).

This idea was present in the manifesto project, with one teacher saying that 'it is important to empower and be hopeful rather than terrifying into inaction' (teacher, *Manifesto* project); young people were in agreement: 'education should empower us to demand change and to demand the rights we should have' (youth, *Manifesto* project), and 'I agree, education should be empowering instead of overwhelming' (youth, *Manifesto* project). There was discussion between teachers as to the place of hope and positivity relative to realism, with certain teachers favouring a 'positive but urgent' framing for EfES while others highlighted the 'risk' that positivity could disempower arguments for change; as one teacher said in response to notions of empowerment related to positivity:

*I'd be interested in hearing more about positivity . . . I don't mean to suggest we need to be negative and doom-laden. But a certain realism is necessary, that can balance the improvements . . . alongside the very real ongoing challenges.* (Teacher, *Manifesto* project)

## 4. Discussion: Implications for Pedagogy

The data demonstrate that emotions are bound up with actions, and this suggests that education may play an important role in transforming environmental emotions. The mechanism for transforming emotions is changing appraisal [30], for example, by creating coping potential or improving future expectations. This means that if educators are to respond to negative emotions associated with environmental sustainability, they must use pedagogies that attend to both primary (the stake one has in the situation) and secondary (options and prospects for coping) appraisals.

The emotions expressed when discussing education for environmental sustainability are largely those that Lazarus [30] identifies as being associated with harm, loss, or threat (anxiety, fear, and guilt) or borderline cases (hope), rather than those associated with benefits. Hope is ambiguous because it can be constructive (action-oriented) or based on denial (inaction-oriented). Table 3 presents our analysis of the emotions found in the dataset, namely, anxiety, fear, guilt, and hope, as well as the associated primary and secondary appraisal questions. We suggest that this analysis might provide a helpful framework for educators when planning and developing climate change education programmes. In our dataset, there was little mention of happiness and joy, pride, gratitude, or love (although love for the planet was expressed). The negative (or at best, ambiguous) emotions were associated with states including isolation, powerlessness, and pressure.

Interestingly, anger did not appear in our dataset. It may be that the group of teachers, teacher educators, and young people participating in educational workshops on environmental sustainability took anger for granted, (i.e., feeling it but not expressing it), or it may be that other emotions were experienced due to the nature of the appraisal made. It is important to note that at the outset of our research the study of emotions was not the focus of our work. Rather, this focus developed as an important theme during our analysis of the data. Had we approached the study with emotions as a guiding research question, other emotions felt by participants, including anger, may have been expressed. Lazarus [30] describes anger as an emotion arising from being slighted or demeaned, where another can be blamed and where someone has full control of the demeaning action. In the context of environmental sustainability and education, it is difficult to attribute full responsibility to a single body, and it may be the case that the participants' understanding of the situation resulted in experiences of anxiety due to the lack of an obvious agent to be held accountable. However, participants identified government, corporations, and educational leaders (among others) as barriers to education for environmental sustainability. Participants indicated that they were able to attribute responsibility; thus, it may be that they saw the possibility of improving the future situation through individual and collective action and as a result hope was more prevalent in this example.

**Table 3.** Analysis of emotions (based on Lazarus [30]).

|  | Appraisal Question | Anxiety | Fear | Guilt | Hope |
|---|---|---|---|---|---|
| **Primary appraisal** | What is at stake and how important is it? | In the context of the climate crisis, the planet and the self and others. | | | |
| | Is the circumstance harmful or threatening or beneficial? | The circumstance is both harmful (in the present) and threatening (now and in the future). | | | |
| | Does the goal at stake preserve or enhance a person's ego identity or moral value? | Existential threat to the self and others. | | | |
| **Secondary appraisal** | Can responsibility be attributed, and to whom? | Responsibility cannot easily be attributed. | Responsibility is attributed to others | Responsibility is attributed to the self. | Responsibility can be attributed to the self and/or others |
| | Can the relationship between the person and the circumstance be improved? | Difficult to determine (because the threat is vague and spatially and temporally complex) | Action tends to avoidance or escape | Action can be taken to improve circumstances but may be difficult and is focused on the individual | Individual and collective action can be taken to improve the circumstances. |
| | Will things work out favourably? | Difficult to determine. | Unlikely | Unlikely | Likely or possible. |

The impact of hope on behaviour depends on whether it is constructive or based on denial [45]. Constructive hope is associated with pro-environmental behaviours and involves positive reappraisal of a situation (e.g., more people being aware of climate change), trust in other actors, and belief that people acting together can make a difference. This is consistent with Ramjam [46], who notes that emotions and actions are important dimensions of youth environmental citizenship. In order to create conditions for constructive hope, Ojala [45] suggests that educators co-create stories about the future with students and avoid extreme cynicism about politicians. The workshops which provided the data for this analysis were designed to bring people (teachers, teacher educators, and young people), usually strangers to each other, together in order to co-create products (a manifesto, youth guide, and policy brief), and featured activities intended to build intimacy and trust between the participants. Avoiding cynicism about politicians is difficult, particularly where the cynicism comes from the participants themselves and avoiding cynicism does not alter the situation. An alternative approach to constructive response is to focus on the processes of decision-making, rather than the decision-makers themselves, and to identify and/or create opportunities and methods for intervening in decisions.

Below, we discuss pedagogies that respond to emotions by changing potential appraisals. These are discussed according to whether they help people to have a stake in education for environmental sustainability, help to attribute responsibility, increase coping potential, and improve future prospects.

*4.1. Having a Stake in Environmental Sustainability*

Environmental damage is harmful and anthropogenic climate change poses risks, including to human health and ways of life. To experience emotion, one must understand what is at stake and see it as important. This is expressed as a primary appraisal, for example, in the question 'what is at stake and how important is it?' (Table 3). As with Lombardi and Sinatra [25], who focused on science teachers' emotional responses to climate change, we found that the emotions we discuss here are present among both teachers and young people. Lombardi and Sinatra [25] argue that more knowledge of content and further reflection on scientific evidence is likely to lead to improved emotional stances and understanding. Our data and our analysis based on Lazarus' appraisal model of emotions [30] suggest that more knowledge and reflection on evidence is in itself unlikely

to change emotional responses. What is needed are pedagogies that can help people care for each other and for the planet. A connection to nature has been associated with enjoyment of nature, empathy for creatures, a sense of oneness, and a sense of responsibility [47], suggesting that education which fosters care for the environment will involve education about the environment.

Creating and maintaining relations of care in education requires listening, dialogue, critical thinking, reflective response, and making thoughtful connections [48]. This implies that the carer (here, the teacher) must be attentive to what the cared-for (the young person) experiences and expresses, even when this is at odds with what the school assumes is important or if it creates conflict with what the school prioritises, e.g., curriculum coverage in the service of examination results. Regardless of whether or not the carer can satisfy the needs of the cared-for, Noddings [48] argues that they must respond in a way that maintains a relationship of trust and keeps communication open. Dialogue is for listening to and responding to the ideas of others, with critical thinking enabling everyone to understand opposing views and respond reflectively and with attention to the other's need for human regard, even when unpersuaded or when the other is wrong. In climate change education, these dimensions of caring are consistent with Rousell and Cutter-Mackenzie-Knowles' [18] call for participatory, interdisciplinary, creative, and affect-driven approaches as well as with calls from young people for education which builds a connection with nature and is based on love for each other and the planet [41,42]. Participants in both the *Manifesto* and *Geoengineering* projects identified the importance of discussion and dialogue in low-stakes educational contexts to create opportunities for tension and challenge to be negotiated.

### 4.2. Identifying and Taking Responsibility

Our findings correspond to those of Hickman et al. [34], who found negative emotions associated with climate change among young people all around the world, particularly emotions attributed to government inaction. This was a common thread among both our teacher and youth participants, and is a secondary appraisal; for example, the question 'Can responsibility be attributed—and to whom?' (Table 3). We heard young people and teachers talk about responsibilities that everyone has, including their peers and colleagues, school leaders, governments, and corporations, pointing to the importance of both individual and collective responsibility.

In terms of collective responsibility and the responsibility of those in power, Hickman et al. [34] argue that protection from mental health problems can come from having feelings heard, validated, respected, and acted on by those in power as well as from pro-environmental action. Providing mechanisms for young people to participate in decision-making are therefore important in schools as well as in wider society. In addition to creating ways for young people to inform decision-making, it is important that they know how decisions are made. This means that education must pay attention to politics, in the sense of Hay [49], that is, as a social activity based on deliberation that happens in situations of choice and where there is capacity for agency. While this might be located in Citizenship classes, it indicates a role for interdisciplinary approaches in subject lessons as well. Climate change and sustainability are often located in science lessons, and in this context, Brock and Glackin [50] examine the responsibilities of teachers when teaching climate change. They identify ways in which the epistemic duty to support students such that their beliefs align with evidence creates tensions with student wellbeing, freedom, and how activism is positioned [50]. There is a need for public dialogue both within classrooms and between teachers and other education stakeholders to release these tensions. This is likely to require leadership from national bodies, as in England at least, many schools are no longer accountable to communities through local democratic bodies, and instead are part of groups of schools or trusts with their own leadership and accountability structures and which are outside of local government frameworks. Environmental, sustainability, and climate education in England tend to fall between the science and geography curriculum [51], and it is important to highlight the limits and limitations of these subjects in dealing with

difficult problems and to examine the ways in which these subjects can help students to take responsibility themselves and demand it of their leaders both in school and in society.

Responsibility is a key theme of the work of Stein et al. [52] in the context of environmental sustainability. The basis of their analysis is that climate change and its associated planetary and human crises, rather than being rooted in ignorance and therefore soluble with more knowledge, represent an ongoing investment in modernist–colonial ways of being which for certain people provide unrestricted and unaccountable autonomy and position responsibility as an individual choice. Stein et al. [52] understand accepting responsibility to mean recognition of how the lives we lead have been and continue to be subsidised by invisibilized exploitation and how we are invested in this system. This means working through "insecurities, projections, fragilities, harmful entitlements and aspirations and desires for certainty, innocence, authority, exceptionalism, and validation" [52] (p. 10) and making a commitment to action that is needed, rather than what is wanted; in short, that is, rejecting individualistic ways of being that consume or withdraw from the world and questioning what it is that should be sustained. Taking responsibility, pedagogically speaking, means making space to examine complexity, uncertainty, complicity, failures, and difficulties and then taking action to overcome feelings of being overwhelmed or powerless. Taking responsibility is a risky act in England, where opposition to capitalism is branded as an extreme stance in the guidance issued to schools [53].

*4.3. Developing Coping Potential to Ensure Positive Futures*

Coping potential refers to the action tendency associated with emotions and the ability to influence the person–environment relationship for the better in order that expectations for the future are improved. Coping potential is a secondary appraisal, for example, the question 'will things work out favourably?' (Table 3). Coping strategies may be problem-focused or emotion-focused; here, we focus on problem-focused strategies in order to deal with the root causes of negative emotions. Sanson, Van Hoorn, and Burke [20] talk about the importance of coping strategies focused on meaning, which involve trust, positive reappraisal, and realistic hope and are useful for challenging problems which cannot be solved immediately. Empowerment and activism can propel young people out of the anxiety and despair they often experience [54].

One way to develop coping potential linked to future expectations is to use futures thinking to identify more desirable ways of being. Futures tools have been used by governments to develop policies [55] as well as in educational practice to examine the future of food and farming [55–57]. In the *Manifesto* project discussed here, we invited participants to critique the present, envision a future of education for environmental sustainability, identify the barriers to implementation, and propose solutions. This technique can be used to generate ideas in response to different social problems [58]; in education, it can be used by teachers, students or school leaders to identify a problem and together envision a more positive future.

While visioning is an important step, it must be followed up by action that makes progress towards making the desired future a reality. This can involve young people and teachers taking part in research in new directions, whether in education (to identify new and more sustainable ways of doing and being in education) or in research. For example, authentic scientific research projects and citizen science projects can provide young people with opportunities to work on genuine problems. In the context of climate change and biodiversity, Rushton [59] identified how research conducted by students in schools can develop students' sense of agency in dealing with global challenges, including climate change and the loss of biodiversity, among students not already active in pro-environmental activity. In addition to student agency, young people and their teachers can make a contribution to knowledge and to society through their work. There is evidence that the kinds of engagement in citizen science which enable lay people to influence the questions that are addressed and which offer people a voice in local decision making can provide a positive contribution to social wellbeing [60]. More broadly, organisations

are beginning to produce educational experiences which are both mindful of emotional responses and include ways to become a changemaker [61]; our research approach was consistent with actions towards influencing the person–environment relationship for the better by co-creating outputs with youth directed at decision-makers, specifically, a policy brief and a manifesto.

The actions taken to develop coping can be carried out either in the environment or for the environment. For example, Walsh and Cordero's [62] report on youth science expertise, environmental identity, and agency in climate action filmmaking, which particularly engaged students who did not have strong identifications with the environment, is an example of education for the environment, as it is directed towards environmental preservation or improvement [63] by enabling students to communicate their climate change solutions and stories with the community. Education in the environment that develops coping potential includes examples which develop environmental concern or pro-environmental behaviour, for instance, appreciative activities such as observing or photographing wildlife [64]. Other ways of developing coping potential include involving young people in environmental and climate action, for example, involving young people in changes to the curriculum and policy, developing their capacity to take action, e.g., through environmental lawsuits, writing letters or other communications, and how to run for public office. The importance of affective pedagogies in responding to both the positive emotions (e.g., hope) and negative emotions (e.g., fear and worry) experienced by children and young people in the context of climate change education has been previously highlighted by researchers [65]. Other pedagogical responses in the context of climate change and sustainability education have included creative [66] and embodied [67] practices as ways of enabling transformative education experiences which promote a shift in consciousness and potentially a change in behaviour [68]. This current research further underlines the need for education to create space for the feelings and emotions which children, young people, and their teachers bring to these issues.

We have outlined the emotions that are expressed by young people, teachers, and teacher educators when discussing education for environmental sustainability, working with the understanding of emotions as appraisals. We did not set out to investigate emotions; rather, these were prevalent in our data as a response to questions about education for environmental sustainability. Emotionally-responsive pedagogies necessarily involve attention to environmental action. The data generated and the means by which they were produced in the manifesto and geoengineering projects indicates the importance of action in developing coping potential, consistent with Rousell and Cutter-Mackenzie-Knowles' [18] call for empowering educational processes; we include participating in research as an example of such an educational process.

We recognise that the study of emotions was not present at the conception of this research and that this focus developed during our analysis of participant contributions. Such an approach may pose a limitation to our work in that there may have been other emotions that were felt by young people, teachers, and teacher educators involved in the *Manifesto* and *Geoengineering* projects which were not expressed due to the research design. We note that this research centres on participants located in the Global North, predominantly in the UK, where the impacts of climate change and ecological crises are felt differently and inequitably compared to those in the Global South, where populations continue to live out the social, cultural, economic, and environmental legacies of imperialist relationships with colonised communities. This limitation was recognised by the participants in both projects, who highlighted the absence of perspectives from beyond Europe and shared their desire to be able to actively listen to and participate in dialogue with those from across the globe. Due to these limitations, we do not seek to claim that this research represents the voice and emotions of all youth. Yet, we argue this research is an important contribution to better understanding emotions as expressed in the context of intergenerational dialogue concerning global climate and ecological crises and understanding possible pedagogical responses.

## 5. Conclusions

Young people, teachers, and teacher educators experience emotions associated with education for environmental sustainability. These tend to be negative (fear, anxiety, guilt) or ambiguous (hope) depending on appraisals of the situation in terms of responsibility, coping potential, and expectations for the future. Transformation of negative emotions into more positive emotions requires a change in how the person–environment situation is appraised. Education has the potential to contribute to the transformation of emotions related to the environment and climate change through emotionally-responsive pedagogies in education for environmental sustainability. These pay attention to the causes of negative emotions by creating agency to act on environmental crises, treating the emotional and other consequences of climate change as well as the causes by taking action which updates and modifies understanding of oneself and the world. Emotionally responsive pedagogies help to identify responsibilities, improve coping potential, and improve future outlook. They can be characterised as caring (involving listening, dialogue, critical thinking, and meaningful connections between the student, their life, and the world), as using knowledge for action, and as involving authentic experiences such as co-creation, authentic research, and outdoor education. For teachers, emotionally-responsive pedagogies represent a way of taking intergenerational responsibility.

**Author Contributions:** L.D. and E.A.C.R. were involved in the research design, data analysis, and in all stages of writing of this paper, project management, and funding acquisition. All authors have read and agreed to the published version of the manuscript.

**Funding:** This research was funded by British Education Research Association Commission 2021–2022 awards made to both authors. The Geoengineering project was funded by an ESRC Impact Acceleration award made by the University of York to Lynda Dunlop (no award numbers available).

**Institutional Review Board Statement:** Institutional ethical approval from the University of York was obtained prior to the commencement of this study (9 March 2020, Reference 20/18).

**Informed Consent Statement:** Informed consent was obtained from all subjects involved in the study.

**Data Availability Statement:** No data are available for this study.

**Acknowledgments:** We would like to thank all of the young people, teachers, and teacher educators who contributed to the *Manifesto* and *Geoengineering* projects.

**Conflicts of Interest:** The authors declare no conflict of interest.

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
