# Peer review of "Education for Environmental Sustainability and the Emotions: Implications for Educational Practice"

_sustainability, doi:10.3390/su14084441_

Round 1

Reviewer 1 Report

This paper explores emotions experienced by young people and teachers in relation to environmental sustainability, in particular climate change, and proposes emotionally-responsive pedagogies for addressing these. It is an extremely interesting paper which has significant potential to be relevant for researchers and practitioners at various levels; you draw out some interesting points which might contribute to broader discussions around pedagogy and practice for environmental and sustainability education. I just have a couple of minor points which it might be useful to address before publication.

In the conceptual framing, I wonder if it would strengthen your paper if you made more reference to slightly more psychology literature in relation to emotional/mental health impacts of climate change. This does not need to be extensive, but this is a burgeoning field and it would be perhaps helpful to show stronger connection to / awareness of it?

Line 46 – when saying sustainability and climate change education is absent, obstructive and diminished in England – would it be relevant to mention the Draft strategy here (it seems rather an omission)?.

Table 1 – Is this in relation to climate change? Might it be helpful to stipulate in table description.

Line 339 – Should this quote be in italics?

Table 3 – I wonder if slightly more explanation of this / linking in the text might strengthen the analysis you are presenting here?

The Discussion and implications for pedagogy section is really interesting and very helpful. I wonder if you might refer more extensively to some of the wider literature around ESE pedagogies, particularly affective or embodied pedagogies to support your argument?

Author Response

Reviewer 1

This paper explores emotions experienced by young people and teachers in relation to environmental sustainability, in particular climate change, and proposes emotionally-responsive pedagogies for addressing these. It is an extremely interesting paper which has significant potential to be relevant for researchers and practitioners at various levels; you draw out some interesting points which might contribute to broader discussions around pedagogy and practice for environmental and sustainability education. I just have a couple of minor points which it might be useful to address before publication.

No response required.

In the conceptual framing, I wonder if it would strengthen your paper if you made more reference to slightly more psychology literature in relation to emotional/mental health impacts of climate change. This does not need to be extensive, but this is a burgeoning field and it would be perhaps helpful to show stronger connection to / awareness of it?

Thank you for this helpful comment. We have added reference to two examples from this literature in the conceptual framework and made reference to three further examples in the introduction (final paragraph).

Line 46 – when saying sustainability and climate change education is absent, obstructive and diminished in England – would it be relevant to mention the Draft strategy here (it seems rather an omission)?

Thank you for this helpful comment. We have added reference to the draft strategy at this point.

Table 1 – Is this in relation to climate change? Might it be helpful to stipulate in table description.

The analysis of emotions, with the example of anxiety, is based on the appraisal tradition. This is not specially in relation to climate change but is a broader consideration of anxiety. We have added a clarification in the paragraph above Table 1.

Line 339 – Should this quote be in italics?

Thank you for highlighting this, we have amended accordingly.

Table 3 – I wonder if slightly more explanation of this / linking in the text might strengthen the analysis you are presenting here?

Thank you for this suggestion. We have made explicit links between Table 3 in each of the sections in the discussion including in the second paragraph of section 4 and in each of sections 4.1, 4.2 and 4.3.

The Discussion and implications for pedagogy section is really interesting and very helpful. I wonder if you might refer more extensively to some of the wider literature around ESE pedagogies, particularly affective or embodied pedagogies to support your argument?

Thank you for this helpful response. We have included an additional paragraph in section 4.3 which makes reference to both affective and embodied pedagogies as suggested by the reviewer.

Reviewer 2 Report

Congratulations for the work. I don´t have to add anything.

Author Response

Reviewer 2

Keywords should be in alphabetical order.

These have been amended accordingly.

In the Introduction the authors use the term "climate change" which I interpret to be mean long-term changes to the earth's climate and weather due to influences from things other than human induced (e.g. the Earth's tilt, the Sun's output, movement through the galactic plane, etc.) . Term 'global warming' on the other hand seems to be used to imply changes to the world's weather and environment due to human induced factors such as the use of cars and coal powerplants. Therefore, if individuals are asked their concerns about climate change, knowing the distinction, they might respond they are not very worried as they might feel there is nothing they can do about how the sun acts. Maybe somewhere in the introduction there should be a discussion about these two ideas and how they authors are using them in their study?

Thank you for this important reminder. We have added a clarification after the first use of the term ‘climate change’ to explain how we are using this term in this study.

In Line 76 and line 77 the authors once again use the term 'climate change'. I think most are stressed about global warming, not climate change. These terms need to be defined better in this study and clarify what is being asked of the sample respondents as the answers to the questionnaire items can be significantly different depending on which term is used.

Thank you for this important reminder. We have added a clarification after the first use of the term ‘climate change’ (page 1) to explain how we are using this term in this study.

I like the term 'environmental sustainability' for use to describe the 'problem'. This then can imply the demands of a growing population and the lack of resources such as water and food. This then removes the hotly debated discussion concerning climate change vs. global warming. Therefore, I am suggesting that the authors use ES as a more neutral term that almost all would agree on needs to be addressed and is a significant issue of importance.

Thank you for this comment. We have clarified our use of the term climate change in response to the previous two comments. We agree that environmental sustainability is a more neutral term however, as climate change is a ubiquitous term we feel it is essential to retain this, but with the important clarification that the reviewer has set out.

In line 110 the authors introduce the term 'geoengineering'. Once again a hotly debated topic which means many things to many people. Therefore, the authors need to go into detail about how they are interpreting this idea or how the 'geoengineering project' defined it.

Thank you for this comment. When we introduce this term at this point in the article, we define geoengineering as ‘technological responses to climate change, based on large-scale human intervention in the Earth’s climate’. We have added a sentence immediately following this definition to ensure it is clear to the reader that this is the definition of geo-engineering which we used in this project and in this paper.

The term 'powerlessness' seems to be a theme that many respondents expressed. With climate change, I would agree with this emotion. However, the whole idea behind geoengineering is to modify the world's climate patterns somehow to offset the resultant cause. I think the authors need to go into more detail about what methods their sample group know about when they hear the term 'geoengineering'. What are the samples used in this discussion/seminars/projects?

We have added some examples of geo-engineering approaches which the participants explored as part of our description of the project and workshops in the research design section. We would contend that powerlessness is an emotion which our participants expressed as part of the geoengineering projects. For example, they expressed a sense of powerlessness in relation to being able to influence global policy around geogeningeering.

In lines 278 through 281 there is discussion about geoengineering and corporate sustainable practices. I see no relationship between the two as geoengineering is taking place at a governmental level or even global level. Once again, the authors need to define how their use of this term is being used and what from of projects/capabilities are being used to implement it. Some might say HAARP is geoengineering. Others contend is being done in US skies with chemtrails. Someone else might even contend that China's south to north canal projects to move water is another form. We need details to better understand and interpret the comments.

In response to the reviewer’s previous two comments we have clarified our shared definition of geo-engineering and included some specific examples of approaches which were discussed as part of the workshops. We suggest that these clarifications respond to the point made by the reviewer here. We also contend that there is a relationship being made by the participants in terms of the approach to sustainable practices and geo-engineering that is one of empowerment rather than shaming.

Line 381-I agree.

No response required.

Lines 462-463. Yes, very important point.

No response required.

Editorial Feedback requesting Ethical Approval Number

The ethical approval reference is included in section 2.1 and at the end of the paper in the Institutional Ethical Board Review statement.

Reviewer 3 Report

  1. Keywords should be in alphabetical order.
  2. In the Introduction the authors use the term "climate change" which I interpret to be mean long-term changes to the earth's climate and weather due to influences from things other than human induced (e.g. the Earth's tilt, the Sun's output, movement through the galactic plane, etc.) . Term 'global warming' on the other hand seems to be used to imply changes to the world's weather and environment due to human induced factors such as the use of cars and coal powerplants. Therefore, if individuals are asked their concerns about climate change, knowing the distinction, they might respond they are not very worried as they might feel there is nothing they can do about how the sun acts. Maybe somewhere in the introduction there should be a discussion about these two ideas and how they authors are using them in their study?
  3. In Line 76 and line 77 the authors once again use the term 'climate change'. I think most are stressed about global warming, not climate change. These terms need to be defined better in this study and clarify what is being asked of the sample respondents as the answers to the questionnaire items can be significantly different depending on which term is used.
  4. I like the term 'environmental sustainability' for use to describe the 'problem'. This then can imply the demands of a growing population and the lack of resources such as water and food. This then removes the hotly debated discussion concerning climate change vs. global warming. Therefore, I am suggesting that the authors use ES as a more neutral term that almost all would agree on needs to be addressed and is a significant issue of importance.
  5. In line 110 the authors introduce the term 'geoengineering'. Once again a hotly debated topic which means many things to many people. Therefore, the authors need to go into detail about how they are interpreting this idea or how the 'geoengineering project' defined it.
  6. The term 'powerlessness' seems to be a theme that many respondents expressed. With climate change, I would agree with this emotion. However, the whole idea behind geoengineering is to modify the world's climate patterns somehow to offset the resultant cause. I think the authors need to go into more detail about what methods their sample group know about when they hear the term 'geoengineering'. What are the samples used in this discussion/seminars/projects?
  7. In lines 278 through 281 there is discussion about geoengineering and corporate sustainable practices. I see no relationship between the two as geoengineering is taking place at a governmental level or even global level. Once again, the authors need to define how their use of this term is being used and what from of projects/capabilities are being used to implement it. Some might say HAARP is geoengineering. Others contend is being done in US skies with chemtrails. Someone else might even contend that China's south to north canal projects to move water is another form. We need details to better understand and interpret the comments.
  8. Line 381-I agree.
  9. Lines 462-463. Yes, very important point.
  10.  

Author Response

Reviewer 3

Keywords should be in alphabetical order.

These have been amended accordingly.

In the Introduction the authors use the term "climate change" which I interpret to be mean long-term changes to the earth's climate and weather due to influences from things other than human induced (e.g. the Earth's tilt, the Sun's output, movement through the galactic plane, etc.) . Term 'global warming' on the other hand seems to be used to imply changes to the world's weather and environment due to human induced factors such as the use of cars and coal powerplants. Therefore, if individuals are asked their concerns about climate change, knowing the distinction, they might respond they are not very worried as they might feel there is nothing they can do about how the sun acts. Maybe somewhere in the introduction there should be a discussion about these two ideas and how they authors are using them in their study?

Thank you for this important reminder. We have added a clarification after the first use of the term ‘climate change’ to explain how we are using this term in this study.

In Line 76 and line 77 the authors once again use the term 'climate change'. I think most are stressed about global warming, not climate change. These terms need to be defined better in this study and clarify what is being asked of the sample respondents as the answers to the questionnaire items can be significantly different depending on which term is used.

Thank you for this important reminder. We have added a clarification after the first use of the term ‘climate change’ (page 1) to explain how we are using this term in this study.

I like the term 'environmental sustainability' for use to describe the 'problem'. This then can imply the demands of a growing population and the lack of resources such as water and food. This then removes the hotly debated discussion concerning climate change vs. global warming. Therefore, I am suggesting that the authors use ES as a more neutral term that almost all would agree on needs to be addressed and is a significant issue of importance.

Thank you for this comment. We have clarified our use of the term climate change in response to the previous two comments. We agree that environmental sustainability is a more neutral term however, as climate change is a ubiquitous term we feel it is essential to retain this, but with the important clarification that the reviewer has set out.

In line 110 the authors introduce the term 'geoengineering'. Once again a hotly debated topic which means many things to many people. Therefore, the authors need to go into detail about how they are interpreting this idea or how the 'geoengineering project' defined it.

Thank you for this comment. When we introduce this term at this point in the article, we define geoengineering as ‘technological responses to climate change, based on large-scale human intervention in the Earth’s climate’. We have added a sentence immediately following this definition to ensure it is clear to the reader that this is the definition of geo-engineering which we used in this project and in this paper.

The term 'powerlessness' seems to be a theme that many respondents expressed. With climate change, I would agree with this emotion. However, the whole idea behind geoengineering is to modify the world's climate patterns somehow to offset the resultant cause. I think the authors need to go into more detail about what methods their sample group know about when they hear the term 'geoengineering'. What are the samples used in this discussion/seminars/projects?

We have added some examples of geo-engineering approaches which the participants explored as part of our description of the project and workshops in the research design section. We would contend that powerlessness is an emotion which our participants expressed as part of the geoengineering projects. For example, they expressed a sense of powerlessness in relation to being able to influence global policy around geogeningeering.

In lines 278 through 281 there is discussion about geoengineering and corporate sustainable practices. I see no relationship between the two as geoengineering is taking place at a governmental level or even global level. Once again, the authors need to define how their use of this term is being used and what from of projects/capabilities are being used to implement it. Some might say HAARP is geoengineering. Others contend is being done in US skies with chemtrails. Someone else might even contend that China's south to north canal projects to move water is another form. We need details to better understand and interpret the comments.

In response to the reviewer’s previous two comments we have clarified our shared definition of geo-engineering and included some specific examples of approaches which were discussed as part of the workshops. We suggest that these clarifications respond to the point made by the reviewer here. We also contend that there is a relationship being made by the participants in terms of the approach to sustainable practices and geo-engineering that is one of empowerment rather than shaming.

Line 381-I agree.

No response required.

Lines 462-463. Yes, very important point.

No response required.

Editorial Feedback requesting Ethical Approval Number

The ethical approval reference is included in section 2.1 and at the end of the paper in the Institutional Ethical Board Review statement.